# Local Control Model of a Human Ventricular Myocyte: An Exploration of Frequency-Dependent Changes and Calcium Sparks

**DOI:** 10.3390/biom13081259

**Published:** 2023-08-17

**Authors:** Jerome Anthony E. Alvarez, M. Saleet Jafri, Aman Ullah

**Affiliations:** 1School of Systems Biology, George Mason University, Fairfax, VA 22030, USA; jalvar7@gmu.edu; 2Center for Biomedical Engineering and Technology, University of Maryland School of Medicine, Baltimore, MD 20201, USA

**Keywords:** calcium, calcium sparks, ventricular myocyte, cardiac cell, ionic currents, RyR, LCC, Ca^2+^, computational modeling, heart, excitation contraction coupling

## Abstract

Calcium (Ca^2+^) sparks are the elementary events of excitation–contraction coupling, yet they are not explicitly represented in human ventricular myocyte models. A stochastic ventricular cardiomyocyte human model that adapts to intracellular Ca^2+^ ([Ca^2+^]_i_) dynamics, spark regulation, and frequency-dependent changes in the form of locally controlled Ca^2+^ release was developed. The 20,000 CRUs in this model are composed of 9 individual LCCs and 49 RyRs that function as couplons. The simulated action potential duration at 1 Hz steady-state pacing is ~0.280 s similar to human ventricular cell recordings. Rate-dependence experiments reveal that APD shortening mechanisms are largely contributed by the L-type calcium channel inactivation, RyR open fraction, and [Ca^2+^]_myo_ concentrations. The dynamic slow-rapid-slow pacing protocol shows that RyR open probability during high pacing frequency (2.5 Hz) switches to an adapted “nonconducting” form of Ca^2+^-dependent transition state. The predicted force was also observed to be increased in high pacing, but the SR Ca^2+^ fractional release was lower due to the smaller difference between diastolic and systolic [Ca^2+^]_SR_. Restitution analysis through the S1S2 protocol and increased LCC Ca^2+^-dependent activation rate show that the duration of LCC opening helps modulate its effects on the APD restitution at different diastolic intervals. Ultimately, a longer duration of calcium sparks was observed in relation to the SR Ca^2+^ loading at high pacing rates. Overall, this study demonstrates the spontaneous Ca^2+^ release events and ion channel responses throughout various stimuli.

## 1. Introduction

Human myocyte models describing calcium dynamics have been first developed for ventricular cells where Ca^2+^ concentration is assumed to be contained in a “common pool” for release. Common pool models have been developed for humans [1,2,3] that rely on previously developed mammalian models such as guinea pig, canine, and rabbit. On the other hand, the stochastic behavior was first described by Cannell et al. in rat heart cell recording wherein the electrically evoked [Ca^2+^]_i_ transient does not rise at a uniform rate within the cell [4], which suggests that graded calcium release events that evoke calcium sparks triggered by local L-type calcium channel currents behave in a stochastic manner. This formulation involved the concept of “local control theory” [5] which asserts that the L-type channel triggers a regenerative cluster of several ryanodine receptor (RyR2) channels, and the probabilistic behavior of this cluster results in a graded response with high amplification of Ca^2+^ entering through the sarcolemma. This theory was further refined into the notion that the opening of an individual LCC located in the transverse tubular (T-tubule) membrane triggers Ca^2+^ release from a small cluster of RyRs located in the closely apposed junctional sarcoplasmic reticulum (SR) membrane in the cardiac dyad. One of the first studies to employ such stochastic behavior and biophysically model the LCC–RyR relationship as a functional release unit in a cardiac dyadic space was formulated by Rice et al. [6], wherein rat cardiac muscle was modeled, consisting of one dihydropyridine receptor (DHPR) and eight RyR channels. This was further emphasized by Greenstein and Winslow’s model of a canine cardiac ventricular myocyte that included 12,500 individual CRUs gating according to the local control of the calcium-induced, calcium-release (CICR) phenomenon [7]. Moreover, there is a stochastic model by Himeno et al. further represented the activation and inactivation kinetics of RyR2 during excitation–contraction coupling [8]. The stochastic openings of individual sarcolemma and SR channels represent a statistical recruitment of Ca^2+^ release events from the SR that conforms to the local control theory.

Calcium handling from intracellular stores and the regulation of spark dynamics are one of the critical endeavors in cardiac electrophysiological studies. Early experimental recordings that involved clinical observations explored the control of calcium release from the sarcoplasmic reticulum (SR) in mammalian cells. These explorations helped formulate mathematical representations of the cardiac cell. In relating these experimental measurements to a human setting, the accessibility and data are extremely limited. Even in one of the first human ventricular cardiac models, by Priebe and Beuckelmann [9], the underlying ionic currents that are not yet sufficiently characterized in human ventricular myocytes are adopted from the action potential model developed by Luo and Rudy [10,11] using guinea pig ventricular cell. Furthermore, current human ventricular cardiomyocyte models in literature are largely dependent on using deterministic or empirical equations that describe CICR mechanisms in cardiac cells. A developed spatially detailed model for the rabbit ventricular myocyte by Nivala and co-workers from the past decade [12] effectively captures Ca^2+^ cycling and coupling to membrane voltage over long time scales. However, limitations of capturing Ca^2+^ spark dynamics existed, such as (a) the Ca^2+^ behavior in the dyadic subspace being at equilibrium, (b) the absence of Ca^2+^ flux into the subspace calcium concentration, and (c) realistic Ca^2+^ sparks not properly being reproduced [13]. Furthermore, the 2011 O’Hara–Rudy model of an undiseased human ventricular myocyte was mainly focused on the ionic channel responses to drug blocking, their rate dependencies, and overall comparisons of normal vs abnormal repolarizations. The O’Hara–Rudy model development is considered to be monumental because it provided detailed experimental recordings of human cardiac tissues. Its recent improvement through Tomek and co-workers (referred to as the ToR-Ord 2020 model) also provided a calibrated version of the O’Hara–Rudy model, including the reformulation of the LCCs, replacement of hERG current model, and ionic-to-whole organ dynamics [14]. Although the advancements of their computational methods were exemplary, the (a) rate-dependent changes of Ca^2+^ propagation through cytoplasm and SR, (b) dynamics of elementary events of calcium release from the SR (known as “calcium sparks”), (c) force–frequency relationships, (d) APD and SR Ca^2+^ rate restitution, and (e) the element of stochasticity in the coupling of LCCs’ RyR2 channels in calcium release were lacking. Therefore, these aforementioned limitations require further investigation.

This study also aims to provide insights to the frequency-dependent changes in Ca^2+^ dynamics and sparks in a human setting. The model presented here also uses mechanisms from the stochastic spatiotemporal model of rat ventricular myocyte from Hoang-Trong and co-workers [13] that overcome limitations of Ca^2+^ fluxes in SR and cytoplasm, along with Ca^2+^ spark behaviors. Recent experimental recordings using human isolated heart donors now provide a basis for Ca^2+^ handling and cardiac AP [15,16,17]. The model presented here is an updated, previously published local-control model of excitation–contraction coupling [13,18,19,20] incorporating several modifications seen in human tissue experiments, and providing additional insights into the overall intracellular Ca^2+^ dynamics and cardiac AP morphology.

## 2. Materials and Methods

### 2.1. Model Formulation and Development

The model equations were developed from a previous local control model of excitation–contraction coupling from guinea pig ventricular myocytes [13,18,19,20] and ion channels observed in published human models. Additionally, the ionic current formulations of I_Ks_, I_Kr_, and I_to_ are adapted from Luo–Rudy [10,11], Ten Tusscher–Panfilov [21], and O’Hara–Rudy models [16]. Furthermore, this current human model also incorporates the Tran–Crampin model for the SERCA2a (sarco- and endoplasmic reticulum Ca^2+^ ATPase isoform 2a) pump [22], and a mathematical model of Ca^2+^ leak from the sarcoplasmic reticulum from Williams et al. [19], which is the first to incorporate realistic CRUs that contain stochastically gating RyRs.

The ionic pumps and currents in the sarcolemma contribute to the overall electrophysiological behavior of a cardiac system (Figure 1). This behavior of a cardiac cell membrane is modeled as a capacitor with variable resistances and batteries which are represented by various ionic currents and pumps, which is described by:(1)$$\frac{dV_{m}}{dt}=-\frac{I_{ion}+I_{stim}}{C_{m}}$$
where V is voltage, C_m_ is cell capacitance per unit surface area, I_ion_ the sum of all transmembrane ionic currents, and I_stim_ is the external stimulus current. The model equation of the sum of all transmembrane ionic mechanisms involved in the action potential adaptation including sodium, calcium, and potassium background currents is given by:(2)$$I_{ion}=I_{Na}+I_{to}+I_{LCC}+I_{Kr}+I_{Ks}+I_{K1}+I_{NCX}+I_{NaK}+I_{nsCa}+I_{PMCA}+I_{bNa}$$
where I_Na_ is sodium current, I_to_ is transient outward potassium (K^+^) current, I_LCC_ is L-type Ca^2+^ channel, I_Kr_ and I_Ks_ are rapid and slow delayed rectifier K^+^ currents, I_NCX_ is the sodium-calcium (Na^+^/Ca^2+^) exchanger, I_NaK_ is Na^+^/K^+^ pump, I_PMCA_ is plasma membrane Ca^2+^ ATPase pump, I_nsCa_ is non-specific Ca^2+^-activated current, and I_bNa_, I_bCa_, and I_bK_ are background sodium, calcium, and potassium currents, respectively.

### 2.2. Calcium-Release Units

Cardiomyocyte CRUs in a cardiac dyad consist of couplons where L-type calcium channels (LCCs) in t-tubules are co-localized with ryanodine receptors (RyR2s; type 2 in cardiac cells) in a junctional SR (JSR) membrane. The 20,000 CRUs in this model are composed of 9 individual LCCs and 49 RyRs. These CRUs are intricately connected through the complex organization of the network SR (NSR) which stores the main intracellular calcium in cardiac muscle cells. Similarly, the CRUs are all connected to the bulk myoplasm without any spatial arrangement of release units. To capture the Ca^2+^ release from CRUs, this model includes 3-state ryanodine receptor mode switching (Figure 2) which incorporates cytosolic calcium-dependent and luminal calcium-dependent gating from Paudel and co-workers [20]. A luminal dependence function modulated RyR open probability to match calcium spark characteristics from Williams et al. [19] and also incorporated minimal adjustments to allosteric coupling energies (Section A.2). The second closed state (C_3_) is the RyR2 adaptive state from changes in [Ca^2+^]_i_.

As another component of the CRU, this model also incorporates a 6-state l-type Ca^2+^ channel which is modeled via two pathways—voltage-dependent or Ca^2+^-dependent inactivation of opening states (Figure 3). Following Hoang-Trong and co-workers [13], C_6_ state was added to the 5-state original model from Sun et al. [23] to accommodate stronger depolarization (from −30 mV to −40 mV). This was formulated in order for all the channels to stay in this state (C_6_) when the cell is at rest. The source of entrant Ca^2+^ to this subspace is the influx of calcium via LCCs and calcium release from SR via RyR2s. When a higher level of Ca^2+^ enters this subspace, it enhances the rate of inactivation of LCC and thereby prevents calcium overload [24].

For this model, we have used a 1:5.6 ratio of LCCs to RyR2s for human cardiac muscle cells as described by Bers and Stiffel [25]. The modeled calcium release is fully stochastic, including Ca^2+^ spark behavior, and reproduces the systolic transient [Ca^2+^]_i_ throughout a propagated action potential. The updated local control Monte Carlo simulation includes 20,000 stochastically gating Ca^2+^-releasing units (CRUs) that open in dyadic cytoplasm subspaces, wherein a dyadic subspace contains a cluster of 9 L-type and 49 RyR2 channels.

### 2.3. L-Type Calcium Channel Permeability

The Goldman–Hodgkin–Katz (GHK) formalism was used [26] to represent nonlinearity in the current–voltage (I-V) curve, the assumption of independent permeation between species, and the expression of constant-field theory (i.e., assumption of a constant electric field along the membrane):(3)$$I_{dhpr}^{\left(i\right)}=N_{open,dhpr}^{\left(i\right)}\frac{P_{dhpr}z_{Ca}^{2}{F^{2}V}_{m}}{RT}\frac{{\left[{Ca}^{2+}\right]}_{ds}^{\left(i\right)}\exp\left(\frac{z_{Ca}FV_{m}}{RT}\right)-\frac{\beta_{o}}{\beta_{1}.{\left[{Ca}^{2+}\right]}_{o}}}{\exp\left(\frac{z_{Ca}{FV}_{m}}{RT}\right)-1}$$
where (i) represents the index of a release site, $N_{open,dhpr}^{\left(i\right)}$ is the number of opening DHPR channels, ${\left[{Ca}^{2+}\right]}_{ds}^{\left(i\right)}$ is the calcium concentration in a dyadic subspace, ${\left[{Ca}^{2+}\right]}_{o}$ is the extracellular calcium concentration, $P_{dhpr}$ is the single channel permeability, $z_{Ca}=2$ is the valence of Ca^2+^ ion, R is the universal gas constant, T is the temperature, and F is the Faraday constant. The partition coefficients are $\beta_{o}=0.341$ and $\beta_{1}=1$ in the case of Ca^2+^ ions [27,28], although other ions can also permeate via LCCs [29]. However, due to the large permeability of Ca^2+^ compared to other ions (e.g., P_Ca_/P_Na_ > 1000), only Ca^2+^ current is modeled.

### 2.4. Modified I_to_

Because most existing models were developed from non-human mammalian tissues, there are limited data on the various currents generating the AP. New findings from the O’Hara–Rudy model using non-diseased human mRNA and protein data made the modeling for different transmural cell types possible. However, differences between human and non-human cell properties affect experimental results and invoke different mechanisms of responses to heart rate changes and to drugs [16]. It was reported that the transient outward current (I_to_), present in human and rabbit atrial cells [30,31,32,33], has been shown to recover from inactivation at least two orders of magnitude faster in humans than in rabbits [1,33,34]. Thus, we also introduce a modification of I_to_ using O’Hara–Rudy activation and ten Tusscher–Panfilov inactivation gates. Furthermore, the modified I_to_ activation gate was decelerated (Equation (4)). This is influenced by the Courtemanche–Ramirez–Nattel (CRN) model, where three activation gates are used (r^3^), causing net activation to be slower and net deactivation to be faster than that of a single gate [1]. The ten Tusscher–Panfilov I_to_ epicardial inactivation time constant is also similar to CRN model [21], and this inactivation gate was further accelerated by adjusting to s^0.7^ with the following equations:(4)$$I_{to}=G_{to}r^{3}s^{0.7}(V_{m}-E_{K})$$
(5)$$r_{\infty }=\frac{1}{1+exp\left(\frac{-\left(V_{m}-14.34\right)}{14.82}\right)}$$
(6)$$\tau_{r}=\frac{1.0515}{\frac{1}{1.2089\times \left(1+exp\left(\frac{-(V_{m}-18.41}{29.38}\right)\right)}+\frac{3.5}{1+exp\left(\frac{V_{m}+100}{29.38}\right)}}$$
(7)$$S\infty =\frac{1}{1+\exp(\frac{V_{m}+20}{5})}$$
(8)$$\tau_{s}=85\times exp\left(\frac{-{\left(V_{m}+45\right)}^{2}}{320}\right)+\frac{5}{1+exp\left(\frac{V_{m}-20}{5}\right)}+3$$
where $V_{m}$ is the membrane potential, $r_{\infty }$ is activation gate with its time constant $\tau_{r}$, $s_{\infty }$ is an inactivation gate with its time constant $\tau_{s}$, and E_K_ is the reversal potential of K^+^.

### 2.5. Numerical Methods

The three states of the RyR2 model and l-type Ca^2+^ channels in each CRU are solved using a patented ultra-fast Monte-Carlo simulation [35] on Fermi C2070 GPUs. The RyR2 channels and LCCs at each release site are modeled as a single stochastic cluster, and each release site is fed with a different sequence of pseudo-random numbers derived from the Saru PRNG algorithm by Afshar [36]. The system of ordinary differential equations comprising the model is solved using the explicit Euler method, and an adaptive timestep (10–100 ns) is utilized for numerical stability and to ensure capturing the fast and stochastic gating of DHPR (dihydropyridine receptor) and RyR2 channels. The simulation uses no-flux boundary conditions based upon the assumption that there was not a significant gradient across cells at the border of the simulation [37]. When the channel fires, a smaller time-step is selected; first to ensure numerical stability, second to limit a maximum 10% of the CRUs to having state changes to occur at a specified time [38,39], wherein two channels undergo state transitions in each timestep < 0.6% of the time [40]. All model initial values, simulation figures, buffering constants, and ion channel conductances are found in Appendix B and in the Appendix A.

### 2.6. Pacing Protocols

Simulations and analyses were performed at 1 Hz for 20 s to achieve steady state unless specified otherwise. Dynamic pacing was conducted on the following: (a) slow-rapid-slow was implemented at 0.5 Hz–2.5 Hz–0.5 Hz pacing for exploring the force–frequency relationship; (b) multiple frequencies were used—0.5, 1, 2, 3, and 4 Hz—for rate-dependence investigations; and (c) S1S2 restitution was assessed at 1 Hz with extra stimuli (S2) applied at specified diastolic intervals (DIs).

### 2.7. Predicted Force

The molecular basis of the force–calcium relation in heart muscle was described by Sun and Irving (see [41] for a review), wherein the co-operative mechanism of Ca^2+^ dependence in force generation was shown to be an intrinsic property of the thin filaments (i.e., actin, tropomyosin, troponin) during contraction. Slight changes in [Ca^2+^]_i_ can directly affect cardiac output, which can produce distinguishing factors in cardiomyocyte defects. This Ca^2+^ dependence of isometric force generation was experimentally observed using demembranated ventricular trabeculae by Sun et al. [42] and accurately described by the Hill equation:(9)$$force=\frac{1}{1+10^{n_{H}(pCa-{pCa}_{50})}}$$
where $n_{H}$ is the Hill coefficient (ranging from 3 to 4), pCa represents peak systolic [Ca^2+^]_myo_ in log scale ($pCa=-{log}_{10}{\left[{Ca}^{2+}\right]}_{myo})$, and ${pCa}_{50}$ corresponding to half-maximum force (ranging 5.5 to 6.0). For this study, a Hill coefficient of 3 and half-maximum force of 6.0 was used in computing the predicted steady state force for each peak [Ca^2+^]_myo_ concentration in the specified log scale (see Appendix B, Table A1).

## 3. Results

### 3.1. Excitation–Contraction Coupling Dynamics: 1 Hz Simulations

The human ventricular myocyte model was tested extensively at 1 Hz for 20 s and achieved steady state during the period 3–5 s of the simulation and onwards (Appendix B, Figure A1). The model predictions agree with the experimental data using human heart tissues seen in several studies in the literature [15,16,43,44,45,46] (Figure 4). AP duration is approximately 0.280 s, which is consistent with values recorded in tissue experiments (~0.270 s, [43]; Figure 4A). The transient calcium ([Ca^2+^]_myo_) or cytoplasmic Ca^2+^ concentration rises to ~0.80 µM from the resting value of ~0.09 µM (Figure 4B). The ryanodine receptor (RyR) open fraction remains at values similar to the peak values of [Ca^2+^]_myo_ as described by Jafri et al. [18] during SR clamp conditions (Figure 4C). On the other hand, SR Ca^2+^ recovery or the “rising phase” where Ca^2+^ is being sequestered back by the SERCA pump into the SR occurs at around 0.5 or 0.6 s (Figure 4D; Appendix B, Appendix A). This was also described by Himeno et al. where SR Ca^2+^ recovery occurs at 0.5–0.8 s [8]. The modified transient outward K^+^ current I_to_ formulation behaves similarly on measured experimental data from the O’Hara–Rudy model [16] using isolated non-diseased human ventricular myocytes at 37 °C (Figure 4E) and exhibits the same shape and amplitude (≤1 µA/cm^2^). Lastly, the L-type calcium channel (I_LCC_; Ca_V_1.2) shape closely resemble experimental recordings from epicardial action potentials [47], and an amplitude between 6 µA/cm^2^ (at rapid rate CL = 0.3 s) and 8 µA/cm^2^ (at slow rate CL = 1 s) as described by Faber et al. [48] (Figure 4F).

### 3.2. Interval–Force Relations and the Force–Frequency Relationship

The widely used slow-rapid-slow pacing protocol was simulated to investigate FFR experimentation in a cardiac cell which consists of three steps. We first (a) simulated 0.5 Hz pacing for stabilizing the system, (b) raised the pacing rate to 2.5 Hz—the heart rate with maximal developed force in humans, as recorded in experimental settings—and (c) restored pacing to 0.5 Hz (Figure 5). For all these three steps, the simulation ran for 45 s for stable output.

Accompanied by the frequent stimulation at 2.5 Hz (i.e., a stimulus applied every 0.4 s; Figure 5A), the steady-state force increases. This increase in cardiac contractility is affected by multiple Ca^2+^–signaling effects at higher pacing: (a) the myoplasm continuously loads Ca^2+^ (Figure 5B) due to the decreased Ca^2+^ sequestration into the SR and outside the cell; (b) because the myoplasm is gaining more Ca^2+^, the intake of Ca^2+^ in the SR via SERCA also increases (Figure 5C); and (c) due to the higher entry of Ca^2+^ via the L-type calcium channels that occurs during faster pacing, the RyR opening decreases (Figure 5D). Furthermore, upon the increase in frequency, there is an initial reduction in SR Ca^2+^ release (Figure 5C at 15–17 s). This is primarily caused by the decreased RyR2 open probability, wherein an increasing RyR2 fraction switches into the adapted state (Figure 5D, red line). Moreover, at high SR Ca^2+^ concentrations (i.e., >1000 µM), the Ca^2+^–dependent transition will further move into an adapted or “nonconducting” state [49] which explains the gradual elevation of the RyR adapted state during the 2.5 Hz pacing (Figure 5D at 15–30 s).

Force depends on the amount of Ca^2+^ bound to troponin as previously described. Although the Ca^2+^ is sequestered from the myoplasm into the SR via SERCA and into the extracellular stores via I_NCX_ and I_PMCA_, there is initially less driving force for this Ca^2+^ transport with the sudden application of rapidly recurring stimulus due to its initial decrease in Ca^2+^ (difference between systolic and diastolic Ca^2+^ at the 15th second in Figure 5B) in the myoplasm (Figure 6A). However, even with less driving force, this effect is eventually compensated for by the increased filling of the SR Ca^2+^ (Figure 6B). This SR Ca^2+^ loading occurs due to the rise in the period of time (i.e., every 0.4 s or 2.5 Hz) in which Ca^2+^ enters via I_LCCs_ and I_NCX_.

In the event of dynamic pacing in Figure 5, the frequency-dependent SR Ca^2+^ changes can also be described by its fractional release, specifically pertaining to the smaller difference between diastolic and systolic SR Ca^2+^ during incomplete SR recovery (Figure 6B). Because of this smaller difference, the SR Ca^2+^ fractional release is also decreased at faster rates. It can be inferred that CICR in high pacing is lower even though the SR Ca^2+^ has already reached higher steady-state concentrations.

### 3.3. APD Rate Dependence and Mechanisms Involved at Higher Pacing Rates

The access to isolated animal cardiac tissues for exploring AP morphology has been increasingly viable for approach, but the availability of isolated human cardiac samples has always been expected to be scarce. However, a relatively large study by Page et al. [15] provided a rate-dependent response in APD using 96 human ventricular trabeculae from 20 human heart donors (Figure 7A) which provides a range of acceptable values for human APD measurements. From their observations, the increase in pacing rate resulted in a decrease in APD—at 1Hz, APD90 is between 0.224~0.471 s and APD50 is between 0.156~0.373 s; at 2 Hz, APD90 is within 0.145~0.347 s and APD50 within 0.089~0.255 s. The model presented here exhibits similar AP duration at simulated frequency settings to other experimental results in the literature (Figure 7B).

It has been long established that AP duration is directly related to the corresponding beating rate (Figure 8A–C). With the increasingly frequent stimulus applied to the cell during higher pacing, peak myoplasmic Ca^2+^ concentration also increases (Figure 8D) due to the growing inability of the sarcoplasmic reticulum Ca^2+^ ATPase pump (SERCA2), sodium-calcium exchange (NCX), and plasmalemmal Ca^2+^ ATPase pump (PMCA) to fully deplete the Ca^2+^ ions of the cell. Due to the increasing myoplasmic Ca^2+^ concentration, peak SR Ca^2+^ content ([Ca^2+^]_nsr_) at diastole also increases up to ~1100 µM at 4 Hz (Figure 8E). This phenomenon is also referred to as SR Ca^2+^ loading. Accompanied by the greater influx of Ca^2+^ into the cell, the peak open probability of RyR2s decreases with high pacing (Figure 8F) ([18,52]; see also [53] for a review) as a consequence of inactivation by higher Ca^2+^ concentrations.

The shape and duration of the action potential was affected by the shortening of the pacing cycle length in early experiments using guinea pig ventricular myocyte [54,55,56]. However, without the presence of I_to_ in guinea pig cells, only I_K_ currents were initially studied [57]. In contrast, AP repolarization in humans involves several types of K^+^ currents in AP phases: I_to_ in phase-one rapid repolarization (called phase-one “notch”) [31]; I_Kr_, I_Ks_, and I_to_ are responsible for phase-two repolarization [11,58,59,60]; and I_K1_ is mainly responsible for phase three [61]. In this study, I_to_ peak density begins to reduce after 1.5 Hz (Figure 8G). Although this transient outward current was found to contribute to phase-two AP repolarization, its contribution to APD shortening is minimal at faster rates. This behavior is supported by a recent study by Johnson et al., wherein I_to_ expression is further diminished at faster pacing rates [62], and it is suggested that repolarizing currents other than I_to_ are involved in APD shortening or prolongation during long-term pacing [63]. In relation to repolarizing K^+^ currents, peak I_Kr_ and I_Ks_ values show great influence in AP repolarization up to 2 Hz, then begin to decrease in expression at faster rates (i.e., >2 Hz; Figure 8H,I). This could be a consequence of two mechanisms: (a) I_Kr_ and I_Ks_ are voltage-dependent channels, and their peak values exhibit the same characteristic as the AP amplitude (i.e., peaks rise up to 2 Hz then decrease thereafter; Figure 8L); (b) the activation and inactivation kinetics of I_Kr_ and I_Ks_—I_Kr_ opens rapidly at the end of phase two and then inactivates slowly [16,64], while I_Ks_ is Ca^2+^-dependent and also exhibits slow deactivation kinetics at faster rates [65]. Hence, the slow deactivation kinetics accumulate in density with increased beats up to 2 Hz.

Furthermore, I_Ks_ is Ca^2+^ dependent—even if peak [Ca^2+^]_i_ is shown to be increasing, the depolarizing current (minimum values of tail current) from I_NCX_ is also greater at high pacing (discussed below; Figure 8K) which helps in Ca^2+^ extrusion from the myoplasm. Therefore, I_Ks_ displays a twofold steeper decline (from 2 to 4 Hz: ~14%; Figure 8H) than I_Kr_ (from 2 to 4 Hz: ~7%; Figure 8I). Moreover, the shortening of APD caused by APD restitution has also been shown experimentally to result from the incomplete deactivation of I_Kr_ and I_Ks_ [66]. On the other hand, however, I_K1_ peak values did not show any change in amplitude or expression (Appendix Figure A4). This is expected because the AP was still able to repolarize at 4 Hz and its additional role as a repolarization reserve [64] was not affected.

The influx of Ca^2+^ into the SR escalates while the entry of Ca^2+^ in each beat via the l-type channel decreases (Figure 8J) as denoted by its minimum values (negative). This behavior is well-observed in the kinetics of l-type calcium channel inactivation, where its trigger could be voltage- or calcium-dependent [67,68]. Elevated intracellular Ca^2+^ concentration near the mouth of the l-type channel acts as negative feedback to Ca^2+^ influx [69]. Its evident decrease in amplitude at high pacing, along with increasing [Ca^2+^]_i_ in faster beats, could eventually result in I_LCC_ termination. The reduction in the LCCs in more frequent CICR leads to the decrease in depolarizing Ca^2+^ in phase two (plateau phase) of the action potential, contributing to APD shortening.

It should be noted, however, that a spontaneous diastolic SR Ca^2+^ release normally activates inward I_NCX_ and Ca^2+^ transport [70]. The extrusion of cytoplasmic Ca^2+^ from the Na^+^–Ca^2+^ exchanger is greater at faster pacing rates (from approximately −0.56 to −1.02 µA/cm^2^; Figure 8K). The exchange of three Na^+^ ions for one Ca^2+^ going through frequent stimulation acts as a greater depolarizing current, bringing a positive charge which also shortens the APD. It has also been established in other studies that the I_NCX_ current varies with the amount of [Na^+^]_i_ in the cytoplasm. At the reversal point of the NCX driving force, AP duration shortens with increasing [Na^+^]_i_. Specifically, at [Na^+^]_i_ ≥ 10,000 µM, the outward NCX current during the plateau phase facilitates cell repolarization, whereas at [Na^+^]_i_ ≤ 5000 µM, it has a depolarizing effect [71]. Ultimately, membrane potential amplitude peaks at ~2.5 Hz and begins to decrease at 3 Hz (Figure 8L). This behavior shares a similar characteristic to the maximum developed systolic force in humans [72] which begins to decrease beyond ~2.5 Hz.

APD shortening in higher pacing is presumably an effect of the decreasing I_LCC_ and RyR2 open probabilities in Figure 8. In order to test if the l-type calcium channel is a primary contributor in APD shortening, we varied the calcium-dependent inactivation rate from the six-state LCC model in Figure 3 (O2 → C4: K24). Overall, the increased inactivation rate exhibits shorter APD90s while the reduced inactivation rate shows longer APD90s (Figure 9A). Lower [Ca^2+^]_myo_ concentration was also observed at an increased LCC inactivation rate (Figure 9B), which could also contribute to a shorter AP duration. Decreased I_LCC_ peak density and higher RyR2 open probability were also observed up to 3 Hz (Figure 9C,D) due to the same conditions of increased LCC inactivation. However, the RyR2 open probability is evidently decreased at 4 Hz during the increased inactivation rate, because RyR2 opening goes further into the adapted state as previously discussed in Figure 5D (see Appendix B, Figure A8 and Figure A9 for LCC and RyR states). In contrast, due to the increased opening of the I_LCC_ in the case of 25% reduced LCC inactivation, more calcium is entering the cell. As previously discussed, RyR clusters can spontaneously close at increased concentrations of Ca^2+^, hence its decreased amplitude under the same conditions (Figure 9D, red line).

### 3.4. Restitution Analyses through S1S2 Protocol

APD restitution describes the relationship between APD and the preceding diastolic interval (DI) [73]. Through the S1–S2 pacing protocol, APD restitution and other Ca^2+^-related mechanisms show adaptive behavior for preserving diastole at different intervals (Figure 10).

The extrasystole between 0.3 and 0.7 s showed a slight increase in membrane potential amplitude (~0.10 mV). APD shortening was also evident with closer proximity to the initial beat at the second stimulus S2 diastolic interval (DI) at 0.02 s (S2 at 0.3 s: APD~0.201 s) as compared with longer intervals (S2 at 0.7 s: APD~0.233 s; Figure 10A). Moreover, normal human myoplasmic Ca^2+^ shows a restitution of the extrasystolic beat, wherein Ca^2+^ transients gradually increase, adapting to longer intervals (Figure 10B). The extra stimuli show less Ca^2+^ release at lower intervals (S2 at 0.3 s), but still recover quickly after release (Figure 10C). As previously described, SR Ca^2+^ recovery occurs between 0.5–0.8 s of initial release through the aid of SERCA, I_PMCA_, and I_NCX_. During shorter intervals, the SR Ca^2+^ does not fully recover; thus, the amount of Ca^2+^ concentration into the myoplasm at this period is also decreased. Lastly, similar behavior was observed in the inward Ca^2+^ current—L-type calcium channel density was reduced primarily due to the lower Ca^2+^ concentration released from the SR at shorter diastolic intervals (Figure 10D).

### 3.5. APD Restitution Mechanisms and Determinants: L-Type Calcium Channel, RyR Open Probability, and I_K_ Currents

Figure 11A–D are the summarized changes in amplitudes from Figure 10. More importantly, it is crucial to pay attention to I_LCC_. This is because the L-type calcium channel is well-established as a major determinant of both APD and [Ca^2+^]_i_ [18,74]. At longer DIs, the l-type calcium channels increase in amplitude (Figure 11D). With the larger opening of the I_LCC_, this effect also allows a greater calcium release from the SR (Figure 11C), wherein a lower value indicates more Ca^2+^ was released (further explained below). Thus, at DIs where the recovery of the I_LCC_ is complete from its preceding DI (approximately >0.5 s; diastolic interval at 0.22 s), the calcium transient amplitude also reaches proper levels at >0.82 µM (Figure 11B).

It was demonstrated by Banyasz et al. [75] that I_NCX_ presents an outward (positive) current at the beginning of the AP, turns into a small inward current at early phase two, then inward (negative) I_NCX_ increases at phases three and four. Specifically, they assert that I_LCC_ is the dominant inward current during AP phases one and two, whereas I_NCX_ is the dominant inward current during phases three and four [75]. The model in this study observes that the inward current of I_NCX_ decreases (becomes less negative) at longer DIs (Figure 11E), meaning that Ca^2+^ extrusion is also decreased. This explains its contribution to longer APD, because Ca^2+^ stays longer inside the cell.

The RyR open probability adapts to the successive and incremental Ca^2+^ elevations. As discussed previously, the RyR2 channel can close from very low or very high Ca^2+^ release from the SR. It is important to note that the SR Ca^2+^ level at rest is at ~1000 µM. At shorter DIs, SR Ca^2+^ release is at a minimum (from 1000 to 875 µM or 12.5% decrease at 0.02 DI), while it is further increased at longer DIs (from 1000 to 842 µM or 15.8% decrease at 0.42 DI). Hence, the RyR open probability is also smaller at shorter DIs, then successively increases at longer intervals (Figure 11F).

Both I_to_ and I_Kr_ show constant peak current magnitude at shorter DIs (~0.02 to 0.17 s) that rise thereafter (Figure 11G,H). However, due to the [Ca^2+^]_i_–dependence of I_Ks_, they showed an early recovery because of the increasing transient calcium (Figure 11I).

Calcium-cycling dynamics evidently have important effects on APD, specifically its mediation through calcium-sensitive currents such as the L-type calcium channel. It was previously suggested by Qu et al. [76,77] that the kinetics of I_LCC_ recovery, rather than its amplitude, should modulate its effects on APD restitution and slope (see Appendix B, Appendix Figure A10 for the restitution slope experimental comparisons). In order to confirm this phenomenon, the calcium-dependent activation rate from the six-state LCC model in Figure 3 (C4 → O2: K42) was varied and each I_LCC_ opening duration was measured. In all cases, APD90s show longer duration on increasing diastolic intervals by the S1S2 restitution protocol (Figure 12A). Moreover, as expected, the APD90 with 10% increased LCC calcium-dependent activation rate exhibited longer APD90s than normal and reduced cases. On the other hand, a similar trend is exhibited by the I_LCC_ opening (in seconds), wherein all cases increase in duration at longer diastolic intervals and the 10% increased calcium-dependent activation rate shows longer opening times (Figure 12B). This indicates that the L-type calcium channel open duration has direct contribution to APD’s S1S2 restitution (see Appendix B, Appendix Figure A11, for I_LCC_ states).

### 3.6. Frequency-Dependent Calcium Spark Behaviors and Characteristics

The next set of simulations explores the behavior and characteristics of Ca^2+^ sparks. A large number of Ca^2+^ sparks are triggered by l-type Ca^2+^ channel opening early in the AP (Figure 13). With the depletion of the SR, there are many RyR openings during the plateau phase but no sparks. During the recovery phase, sparks re-appear because the SR Ca^2+^ has started to refill to levels sufficient to maintain CICR. This recovery is evident in the increasing Ca^2+^ spark amplitude. Diastolic sparks are observed at the right of Figure 13 after the end of the AP.

It was previously described by Guo et al. that CICR local control is governed by SR Ca^2+^ load because it determines the single RyR current amplitude that drives inter-RyR CICR [78]. In their study using male rabbit hearts, the observed spark frequency increases with SR Ca^2+^ load because spontaneous RyR openings at high loads produce larger currents or larger CICR “trigger” signals. Figure 5C and Figure 8E demonstrated that [Ca^2+^]_nsr_ concentration rises (or loads) in higher pacing frequencies. Figure 14A shows that the spark duration also increases with pacing rate. With incomplete time for recovery SR Ca^2+^ due to frequent stimulations (1–4 Hz), the average spark peak declines over the course of an action potential (Figure 14B). Additionally, accompanied by an increased influx of Ca^2+^ through the LCCs at high pacing, the I_LCC_ current showed a decrease in amplitude previously described in Figure 8J. In the cardiac muscle, the Ca^2+^ influx through the l-type calcium channels is initiated by calcium sparks which are localized in intracellular Ca^2+^ concentrations near the inner mouth of the channel pore [79]. The decrease in l-type calcium current results in less triggering of calcium sparks at faster pacing beats (Figure 14C). The elevated intracellular Ca^2+^ at faster rates, as previously described in Figure 8D, triggers channel inactivation and provides negative feedback to Ca^2+^ influx [69]. On the other hand, it has also been observed experimentally that the frequency of sparks increases per second (Figure 14D) with SR Ca^2+^ loading [80].

The intrinsic rhythm behaviors of Ca^2+^ sparks are also crucial in the overall Ca^2+^ homeostasis—a finite time is required for local [Ca^2+^]_i_ to decline and reuptake into the SR. This phenomenon could also induce another spark, requiring RyR channels some recovery time after an initial release, and with cellular Ca^2+^ overload, myocytes can also exhibit regular and stable Ca^2+^ oscillations that are independent of the membrane potential [70]. SR Ca^2+^ uptake can increase the resting Ca^2+^ spark frequency, which also increases the depolarization rate during diastole. Satoh and co-workers [81] have stated that the increased SR Ca^2+^ content causes this observed increase in Ca^2+^ spark frequency. They also described the response of SR Ca^2+^ release channel to [Ca^2+^]_i_, which is not static, but depends on the rate and history of [Ca^2+^]_i_—rapid increases in Ca^2+^ are much more effective at opening the Ca^2+^ release channel. This phenomenon is captured by the model in this study, where a sudden increase in Ca^2+^ spark frequency was observed after 3 Hz (Figure 14E). In the Ca^2+^ spark measurements in non-failing human ventricular cells from terminal patients [82], it was observed that the time to peak of Ca^2+^ sparks increases further with a higher Ca^2+^ load (Figure 14F).

## 4. Discussion

### 4.1. Advantages of the Model versus Early Studies

Computational modeling of the heart has been used to understand the complex interactions between the membrane potential, duration, calcium dynamics, ionic currents, and pumps in the ventricular myocyte. The first human ventricular model from Priebe and Beuckelmann was able to display the I_LCC_ and K^+^ currents and the sodium–calcium exchanger using human data, and other currents from the development of the Luo–Rudy phase-two guinea pig model [9]. Ten Tusscher et al. included the differences between endocardial, epicardial, and M cell types [21]. The widely popular O’Hara–Rudy undiseased human cardiac ventricular action potential meticulously described physiological changes to AP, the rate-dependence of various ionic currents, and each respective drug block [16]. A hybrid ventricular model by Himeno et al. reproduced the regenerative activation and termination of the CICR phenomenon, separated the RyR2 inactivated state, and included the experimental measurements of the transient rise in Ca^2+^ concentrations, focusing on the excitation–contraction coupling properties [8]. The recent and latest model by Tomek et al. is a developed version of the O’Hara–Rudy model and includes several validations through drug-block responses, a reformulation of the l-type calcium current and replacement of hERG model [14]. However, most of the aforementioned human models were not able to demonstrate the stochastic nature of Ca^2+^ propagation through the CRUs and relied predominantly on deterministic approaches. For example, the rate-dependent changes of Ca^2+^ propagation through cytoplasm and SR, including the effects of l-type activation and inactivation rates through LCC’s six-state transitions, can be described in this model. Moreover, realistic Ca^2+^ spark dynamics which were not previously described relative to human AP characteristics are now detailed in this study. The force–frequency relationship during dynamic pacing indicative of predicted force and SR Ca^2+^ fractional release was not formerly reported as well, although these are important properties in distinguishing arrhythmic propensities [83]. Appendix A compares the features and capabilities of different models for the human ventricular myocyte. This study hopes to bring new insights to these perspectives in the development of human ventricular cardiomyocyte models.

### 4.2. Interval-Force Relations Depend on RyR Dynamics

Force–frequency relationship (FFR) explores the typical response to a transient increase in stimulation frequency in a ventricular muscle. The relationship between stimulation pattern and contractile force was first investigated by the early works of Bowditch in 1871 [83,84]. A well-established observation during FFR experiments demonstrated that the increase in SR Ca^2+^ load at high pacing frequencies is responsible for the positive Ca^2+^-frequency relation [85,86]. In slow-rapid-slow pacing, two important Ca^2+^-transport mechanisms were involved in the increased myocardial steady-state force: (a) the myoplasm is continuously saturated by Ca^2+^ due to the insufficient sequestration of Ca^2+^ by SERCA, I_NCX_, and I_PMCA_ back into the SR and outside the cell; and (b) because peak systolic myoplasm Ca^2+^ increases, the intake of Ca^2+^ into the SR also increases (SR Ca^2+^ loading). After a few succeeding beats, Ca^2+^ levels in both myoplasm and SR achieve a new steady state. This study also demonstrates the RyR2 adaptability during a Bowditch phenomenon where the RyR2 open probability acclimates to changes in pacing frequency, as denoted by the accommodation of the adapted state in Figure 5D. This phenomenon was first described by Györke and Fill, wherein cardiac RyR appears to adapt to the SR Ca^2+^, preserving its capacity to manage a new higher Ca^2+^ concentration [87]. Moreover, the dynamic pacing also reveals the micro-adjustments of systolic and diastolic SR Ca^2+^ described by its fractional release illustrated in Figure 6B. These multiple synchronizations between calcium-regulatory proteins and pumps are necessary in the observance of healthy cardiac contractility and performance which have not been demonstrated before in cardiomyocyte modeling.

### 4.3. Increased Predicted Force Is Accompanied by Reduced SR Ca^2+^ Fractional Release during Rapid Pacing

SR Ca^2+^ release is rarely measured directly and is most often assumed to be proportional to the measured [Ca^2+^]_i_ transient peak [88]. [Ca^2+^]_i_ binds to troponin, resulting in the sliding of the thick and thin filaments and the development of pressure within the ventricle [89]. It was previously described by Bassani et al. that the effect of SR Ca^2+^ load on fractional SR Ca^2+^ release may display a relationship to the regulation of the Ca^2+^ release channel, meaning that the increased intra-SR Ca^2+^ increases the sensitivity of the SR Ca^2+^ release channel to a given cytosolic Ca^2+^ trigger [90]. This phenomenon has been observed experimentally in ferret and rat ventricular cardiomyocytes, wherein fractional SR Ca^2+^ release also increases SR Ca^2+^ load [90,91]. However, this increase in fractional SR Ca^2+^ release with higher SR Ca^2+^ load was conducted by the application of caffeine or by manually increasing the extracellular calcium ([Ca^2+^]_o_) available to be released, specifically during varied loads with altering [Ca^2+^]_o_ concentrations from 500 µM to 8 mM (see [90] for a review of methodology). Early studies exploring SR Ca^2+^ release suggest that this Ca^2+^ release is proportional to the measured [Ca^2+^]_i_ amplitude. Predicted steady state force depends on the amount of Ca^2+^ bound to troponin, and this can be measured through a modified Hill Equation (9). There is initially less driving force for the Ca^2+^ sequestration back into the SR via SERCA and Ca^2+^ extrusion out of the cell mainly caused by I_NCX_ upon the sudden application of rapid stimulus (from 0.5 to 2.5 Hz as demonstrated). Moreover, the release of SR Ca^2+^ in cardiac muscle during excitation–contraction coupling is known to be graded by the amount of activating calcium outside the SR, which is defined as a calcium-induced, calcium-release event [90]. In rabbit left ventricular epicardium recordings, the difference between diastolic and systolic SR Ca^2+^ exhibits a frequency-dependent response, and diastolic SR Ca^2+^ increases quickly to a new steady state as the pacing rate also increases [92]. From the smaller difference between diastolic and systolic SR Ca^2+^ during incomplete SR recovery discussed in Figure 6B, SR Ca^2+^ fractional release is observed to be decreased at faster rates.

### 4.4. Rate-Dependent Changes and Mechanisms in APD Shortening

Spontaneous RyR openings at high loads produce larger Ca^2+^ currents as a form of CICR trigger signal [78]. Along with an increasing SR Ca^2+^ concentration, this also results in a developing force represented by the increase in transient calcium ([Ca^2+^]_i_) amplitude. However, as Ca^2+^ levels in the cytoplasm rise, Ca^2+^ can trigger the closing of the RyR [52]. Furthermore, the open probability of a RyR channel as a function of Ca^2+^ concentration is revealed to have a bell-shaped curve [93], which represents that this channel can close at very low or very high concentrations due to the Ca^2+^ release from the SR. During CICR, the small influx of Ca^2+^ from the I_LCC_ induces a Ca^2+^ release from the SR through the RyR2s which raises myoplasmic calcium ([Ca^2+^]_myo_) that depolarizes the cell. By observing the APD at faster rates, multiple ionic currents can be accounted for as a mechanism for APD shortening. As described in Figure 9, the L-type calcium channel’s activation and/or inactivation rate could be considered a major contributor to this phenomenon. Rapid pacing shows that multiple mechanisms are involved in the ability of Ca^2+^ to be removed from the myoplasm due to the frequent stimulation, which is generally accompanied by APD shortening. [Ca^2+^]_myo_ and SR Ca^2+^ both increase at higher pacing rates. Along with the influx of Ca^2+^ into the cell, RyR open probability decreases as a consequence of inactivation from higher Ca^2+^ concentrations. Moreover, repolarizing K^+^ currents (I_to_, I_Kr_, and I_Ks_) also show rate-dependent changes and contribute to the APD shortening. An exception to this phenomenon is I_K1_, where its amplitude remained constant from 0.5 Hz to 4 Hz. Furthermore, I_Ks_ also showed [Ca^2+^]_i_ dependence, where it showed a twofold steeper decline than I_Kr_ from 2 Hz to 4 Hz pacing. This steep decline of I_Ks_ may be affected by the greater extrusion of [Ca^2+^]_i_ by the depolarizing tail current of I_NCX_ at the same increasing rates. Lastly, the L-type calcium channel decreases at rapid pacing due to its voltage- or calcium-dependent inactivation [67,68]. These behaviors imply clinical importance. For example, the recovery from inactivation seen in mammalian cardiac fibers [94] occurs more slowly at depolarized potentials, which suggests that APD shortening may be important in preventing heart block during periods of sustained tachycardia as described in early studies of guinea pig and human ventricles [95].

### 4.5. S1S2 Restitution Mechanisms and Determinants

The shape and behavior of AP restitution show similar characteristics to experimentally measured myocytes from human left ventricle by Näbauer et al. [44]. Relative to changes in [Ca^2+^]_i_, its effects are also propagated to other calcium-dependent currents, such as I_LCC_, I_NCX_, and, minimally, I_Ks_, which have direct effects on the duration of the action potential. Our analysis uses this model in predicting how ionic currents and parameters in cardiac models, such as RyR2 refractoriness, I_NCX_ behavior in Ca^2+^ extrusion from the myoplasm, I_LCC_ density, SR Ca^2+^ load, and K^+^ repolarizing currents give rise to APD restitution and mechanical restitution ([Ca^2+^]_i_). It was previously described by Qu et al. [76] that APD restitution at short DIs is dominated by the restitution of the I_LCC_, whereas I_K_ restitution only assumes importance at longer diastolic intervals. Exploring these effects in restitution intervals, this study adds that APD shortening at short DIs can also be contributed to by Ca^2+^ extrusion by I_NCX_ and a low RyR open probability due to the low release of Ca^2+^ by the SR. These analyses are crucial because experimental and simulation studies have shown that restitution of the cardiac APD plays a major role in predisposing to ventricular tachycardia and fibrillation. In mitigation, the electrical restitution of the APD in ventricular muscle has been shown to be a key factor regulating dynamic instability [96]. Respective electrical restitution curves in S1S2 experiments display slow kinetics—less steep restitution curves in humans—during the application of S2 stimulus in low diastolic intervals. Regional variation in APD is known to play a key role in reentrant arrhythmias [97] and the heterogeneous organization of restitution may provide a substrate for arrhythmia [98]. This behavior is a recent experimental finding of Lovas et al., wherein human ventricular APD restitution evidently differs from other mammalian species, such as rats, guinea pigs, rabbits, and dogs—human ventricles exhibit prominent phase 1 repolarization due to a higher level of I_to_ expression, and this is presumed to be associated with the slow restitution kinetics [50].

### 4.6. Calcium Spark Characteristics Change with Pacing

Ca^2+^ sparks are discretized calcium release events due to random and collective openings of the RyR2 channels clustered in a CRU. On a fundamental level, these sparks result from spontaneous openings of single SR calcium-release channels, which are supported by ryanodine-dependent changes of spark kinetics [79]. The RyR is both the SR Ca^2+^ release channel and a scaffolding protein that localizes key regulatory proteins such as calmodulin, calsequestrin, and other proteins at the luminal SR surface [53]. Local CRU Ca^2+^ sparks were investigated using the mathematical model of Williams et al. [19] which includes spatial nanodomain determinants of individual CRU organization and a realistic number of 20,000 CRUs. Systolic transient [Ca^2+^]_i_ activates L-type Ca^2+^ channels at the surface membrane and at transverse tubules which then elevates [Ca^2+^]_i_ locally in the dyadic subspace compartment between the t-tubular and terminal SR membranes. This situation is amplified when RyR2 clusters are activated by locally elevated subspace [Ca^2+^]_ds_ during CICR (see [99] for the physiological review). This spontaneous local increase in intracellular calcium concentration from the SR produces calcium sparks (Figure 13). Moreover, as previously described, SERCA sequesters Ca^2+^ back into the SR storage. During resting SERCA activity, an “SR Ca^2+^ leak” or Ca^2+^ efflux from the SR is present in late phase of Ca^2+^ release during AP, and this leak has been proposed to take the form of spontaneous Ca^2+^ spark activity [100].

Spontaneous local increase in intracellular calcium concentration from the SR produces calcium sparks. It was observed that spark duration also increases due to higher SR Ca^2+^ load because the cell becomes saturated with increasing Ca^2+^ concentration. This phenomenon has been observed experimentally, where the rise in frequency of sparks corresponds to SR Ca^2+^ loading [80]. As previously described, RyR2 open probability decreases with higher SR Ca^2+^ load due to the behavior of RyRs, which can spontaneously close upon increased Ca^2+^ concentrations. This study also reveals that the average calcium spark peak declines at higher pacing due to the spontaneous decay of active RyR2 cluster channels, which has been observed experimentally [78]. Lastly, the decrease in L-type calcium channel activity can be attributed to the fewer triggered calcium sparks in between faster pacing beats, because of the negative feedback property from elevated Ca^2+^ concentrations [69].

### 4.7. Implications of Frequency-Dependent Changes in Ca^2+^ Mishandling

A fundamental property of cardiac myocytes is the observed decrease in action potential duration with an increase in heart pacing rate [15]. The action potential is directly influenced by the sudden and transient depolarization of the excitable cardiac cells. The amount of time between each excitation and relaxation of these cells could also vary, wherein the mishandling of Ca^2+^ propagation in each beat is one of the many pro-arrhythmic factors in cardiac diseases. Additionally, graded fluxes of Ca^2+^ in the cytoplasm and SR are studied in the development of contractile force in which they exhibit highly nonlinear relations [53]. Under the conditions of increased heart rate in healthy subjects, or an increased frequency of stimulation in myocardial cells, they were suggested to result from an increased amount of activator Ca^2+^ released from the SR. As the frequency of stimulation and the force of contraction are increased, there is a correlative increase in Ca^2+^ concentration in the cytosol with each beat [101].

Bers and co-workers suggested that direct measurements of SR free Ca^2+^ concentration provide values of 1.0 to 1.5 mM at the end of diastole, which is only partially depleted (24% to 63%) during contraction [102]. This leads one to believe that cardiac SR leaves substantial Ca^2+^ reserves. Furthermore, a need for adequate measurement of SR Ca^2+^ content was also suggested in relation to the increase in cytoplasmic Ca^2+^ concentration [89]. This is due to the Ca^2+^-sensitive indicators in the SR, which do not seem to provide consistent results among mammalian tissues [103], and the application of caffeine could also reduce the sensitivity of measurements. In vivo, Bers and co-workers also determined maximum force in an intact rabbit myocyte by adding 1 μM isoproterenol [102] which demonstrated an increase in SR Ca^2+^ content within approximately 1 to 1.2 mM. Thus, it is important for in silico simulation studies to properly monitor frequency changes in SR Ca^2+^ loading that fall within experimental observations.

The heart is capable of rapid adaptation and it protects itself against contractile dysfunction during ischemic episodes and reperfusion [104]. This adaptation during rapid pacing also protects the myocardium against the infarction, as seen in pig ventricles by nonischemic K^+^ channel activation [105]. In the exploration of the force–frequency relationship between the mechanisms of a cardiac cell, pacing techniques were able to be understood and used for mitigating cardiac abnormalities. For example, the technique of rapid ventricular pacing has been proven to be effective in the incidence of ventricular arrhythmias, alternative interventional procedures, and even in cerebral aneurysm surgeries [106,107,108]. Observing frequency-dependent behaviors of the cardiac cell are critical to discovering more alternative routes as opposed to interventional means.

### 4.8. Model Limitations

The model presented here was rigorously tested for system stability, force–frequency intervals, Ca^2+^ restitution, and reproducing pathophysiological phenomena. These are important prerequisites for studying the mechanisms of cardiac arrhythmias in humans and simulating drug interventions from a cellular perspective. However, our model does not contain Ca^2+^/calmodulin-dependent protein kinase II (CaMKII) signaling pathways. It was observed in canine myocytes that CaMKII extends the range of transient Ca^2+^ and APD alternans to slower frequencies and increases the alternans amplitude [109]. It also has a role in chloride handling [3,16], which is also not included in the set of ion concentrations in this model. Canine ventricular data were also used in modeling CaMKII, which induced positive rate-dependence on transient Ca^2+^ [110], but only acts as a determinant in [Ca^2+^]_i_, not of APD. As described by the O’Hara–Rudy model, the integrated electrophysiological consequences of CaMKII effects on target channels are minimal, and its suppression had only minor effects on APD rate dependence and AP morphology [16].

An additional limitation of the model is that although the model can generate EADs, it cannot currently produce large amplitude DADs (delayed afterdepolarizations) under Ca^2+^ overload conditions. DADs can be observed when additional changes to the state of the ventricular myocyte are necessary, such as imposing heart-failure conditions, as shown previously by Ullah et al. [37,111]. The reason for this is that there is no spatial organization of the CRUs in the model, where the activation of a release unit can propagate and activate adjacent CRUs. In the current model, each of the CRUs is connected to the bulk myoplasm, causing the effect of a spontaneous release event from a CRU to be diluted. Improving this is the subject of a future work.

Each computational model is commonly validated with data from a subset of samples, which can be animals or humans, collected during in vitro, ex vivo or in vivo experiments [112]. However, it is often difficult to generalize all cardiomyocyte mechanisms to be consistent at a certain threshold. For example, an increased frequency and prolonged Ca^2+^ sparks from an experimental setting were reported to be temperature-dependent, wherein a reduction from 35 to 10 °C was noted to be an important thermodynamic determinant in observing this phenomenon [113]. Moreover, ion channels are distinctively expressed and have different compositions throughout species (e.g., humans, rabbits, guinea pigs, canines, rats, and others) which could affect experimental findings. Subtle electrophysiological differences between species may lead to different rhythmic or arrhythmic cellular behaviors and drug response [114]. Overall, “All models are imperfect, but they are nonetheless useful” by Sutanto et al. [112] is an appropriate statement for capturing the limitations of in silico cardiac models.

## 5. Conclusions

This study developed a local control, stochastic computational model for excitation–contraction coupling in the human ventricular cell in a formulation that included the following components: ionic currents present in humans, the stochastic behavior of RyR2s and LCCs in “open and closed” states, Ca^2+^ fluxes across the sarcolemma, and others. This study was able to provide mechanistic into frequency-dependent behaviors in Ca^2+^ dynamics, S1S2 restitution determinants, and Ca^2+^ spark characteristics that affect the overall AP morphology in a human cardiac ventricular myocyte. The model predicted irregularities in Ca^2+^ dynamics that can help predict such irregularities at the Ca^2+^ spark level, and distinguished abnormal Ca^2+^ dynamics that contribute to the discovery of arrhythmogenic currents. The improvement in the availability of human experimental data makes it possible for these models to continue to be useful in mitigating and predicting cardiac abnormalities.

## Figures and Tables

**Figure 1 biomolecules-13-01259-f001:**
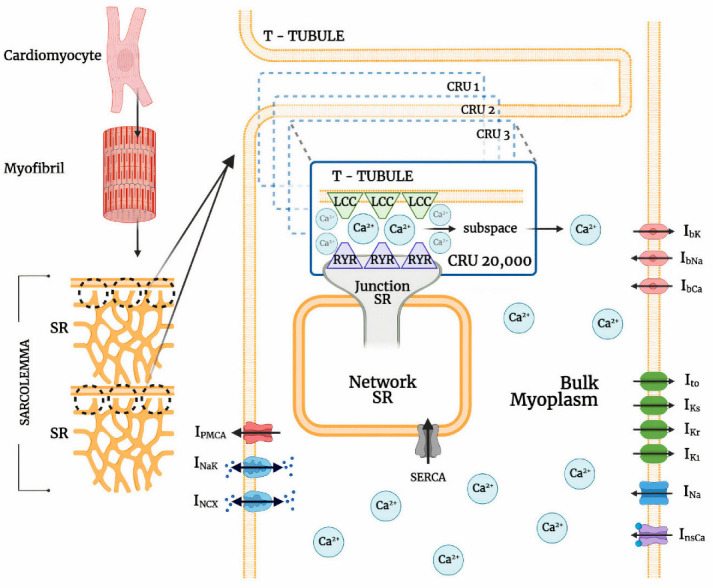
Schematic diagram of a ventricular cell containing 1 transverse tubule (T-tubule) branch with a portion of the 20,000 stochastically gating CRUs. Abbreviations: LCC–L-type calcium channel; RyR2–ryanodine receptor type 2; I_NCX_–sodium-calcium exchange; I_NaK_–sodium-potassium ATPase; I_PMCA_–plasmalemmal calcium ATPase; CRU–calcium release units; SERCA–sarcoplasmic and endoplasmic reticulum calcium ATPase; SR–sarcoplasmic reticulum; I_nsCa_–nonspecific calcium-activated current; I_Na_–sodium current; I_K1_–inward rectifier potassium current; I_Kr_–rapid delayed rectifying potassium current; I_Ks_–slow delayed rectifying potassium current; I_to_–transient outward potassium current; I_bCa_–background calcium current; I_bNa_–background sodium current; I_bK_–background potassium current. Schematic diagram was created through BioRender.com.

**Figure 2 biomolecules-13-01259-f002:**
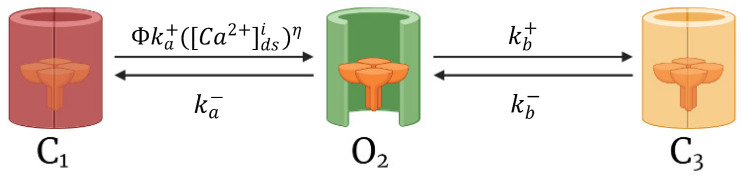
The 3-state ryanodine receptor model. Opening probability (P_o_) of RyR2 channels from closed state (C_1_) to open state (O_2_), is controlled by a luminal regulation function (ϕ). In the resting phase, all RyR2s stay in the closed state (C_1_). Upon the influx of Ca^2+^ in the dyadic subspace, the channels activate into the open state (O_2_), and intermittently, the channels might inactivate into an adaptive state (C_3_). Schematic diagram was created through BioRender.com.

**Figure 3 biomolecules-13-01259-f003:**
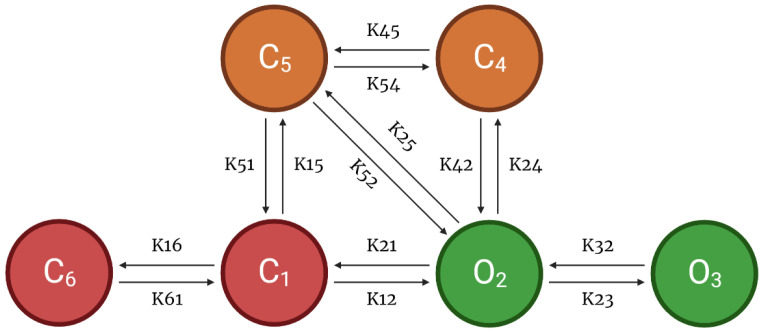
6-State L-Type Calcium Channel Model. This 6-state Markov model is controlled through the voltage-dependent inactivation (O_2_ → C_5_) and Ca^2+^-dependent inactivation (O_2_ → C_4_) by the subspace Ca^2+^ level at each release site. During resting potential, all L-type channels are in a closed state (C_1_). The change in the membrane potential activates the LCCs into an open state (O_2_). Channel in O2 state may continue to open state (O_3_), but the change in voltage brings LCCs into inactivated state (C_5_) or excess Ca^2+^ in dyadic subspace may also transition into another inactivated state (C_4_). Schematic diagram was created through BioRender.com.

**Figure 4 biomolecules-13-01259-f004:**
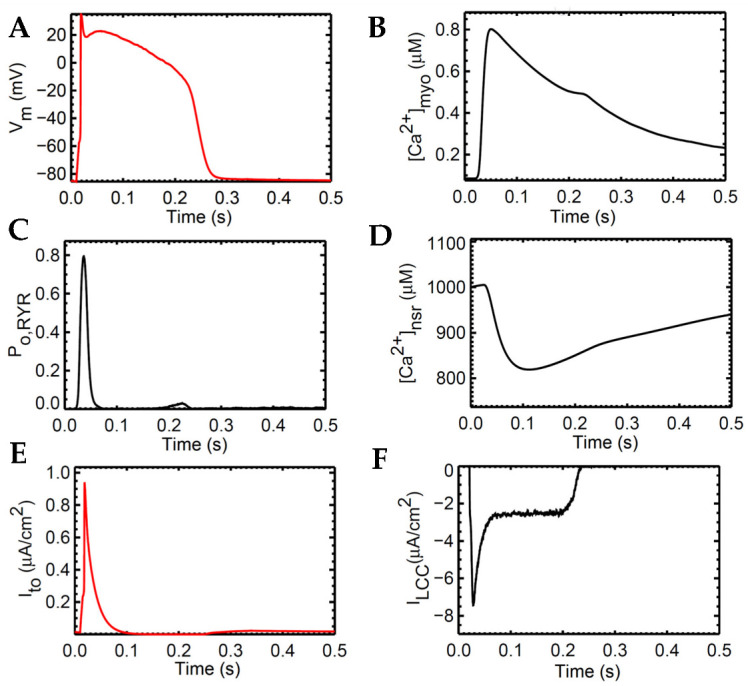
Several 1 Hz baseline simulations. (**A**) APD ~0.280 s. (**B**) Transient calcium ([Ca^2+^]_myo_) representing the increase and reduction in cytosolic Ca^2+^. (**C**) RyR open probability rises during the phase of cell depolarization. (**D**) SR Ca^2+^ recovery from base concentration of 1000 µM. (**E**) Modified transient outward K^+^ current (I_to_) amplitude ≤ 1 µA/cm^2^ which is similar to tissue experiments. (**F**) I_LCC_ shape and amplitude closely resemble experimental recordings from epicardial action potentials.

**Figure 5 biomolecules-13-01259-f005:**
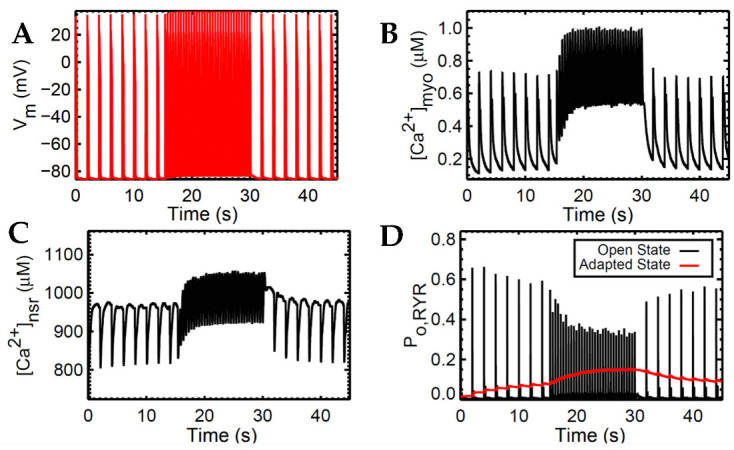
Interval-force relations. Simulation was conducted at 0.5 Hz–2.5 Hz–0.5 Hz pacing. (**A**) AP shows absence of early afterdepolarizations (EADs). (**B**) A few smaller Ca^2+^ transients ([Ca^2+^]_myo_) at the beginning of faster pacing form a “Bowditch” or positive staircase effect then return to a steady state. (**C**) The NSR Ca^2+^ load gradually increases in the rapid pacing (2.5 Hz) exhibiting similar positive staircase effect, then decreases gradually at the beginning of slow pacing (0.5 Hz). (**D**) The peak RyR open fraction (open state) is lower during fast pacing; adapted state (red).

**Figure 6 biomolecules-13-01259-f006:**
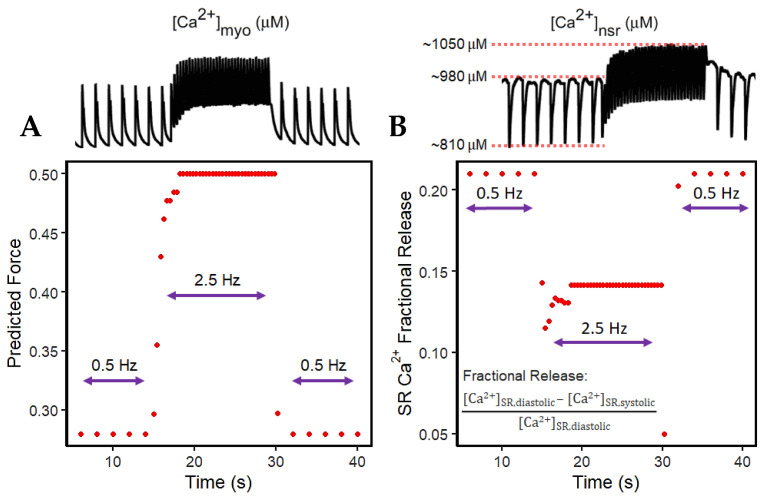
Predicted force and SR fractional release. (**A**) Predicted force from the sudden increase in pacing (2.5 Hz) starting at 15 s using the Hill equation (2.9) using peak systolic [Ca^2+^]_myo_. (**B**) SR Ca^2+^ fractional release is decreased at higher pacing (2.5 Hz) due to the small difference between diastolic and systolic [Ca^2+^]_nsr_ concentrations.

**Figure 7 biomolecules-13-01259-f007:**
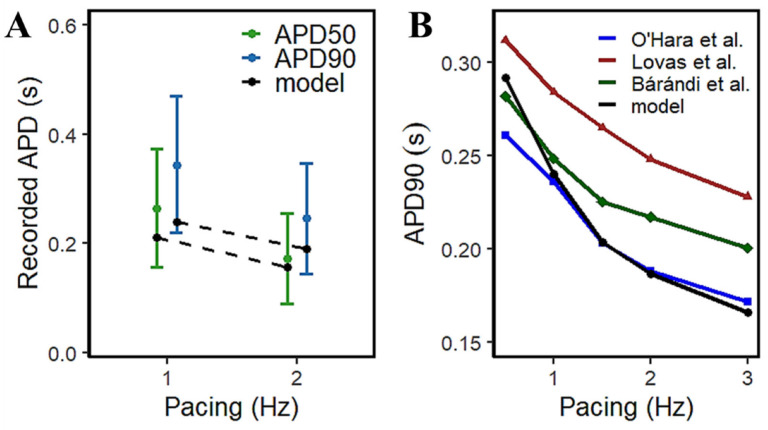
APD90 restitution is similar to experimental recordings. (**A**) Experimental recordings of the large study by Page et al. using 96 human trabeculae which provide a range of human AP duration. (**B**) Other APD90 recordings compared with our model: Lovas et al. (right human ventricular muscle preparations) [50]; O’Hara et al. (measured from scaled expression ratios from multiple recordings for human ventricular epicardial cell types) [16]; Bárándi et al. (human ventricular papillary muscles) [51].

**Figure 8 biomolecules-13-01259-f008:**
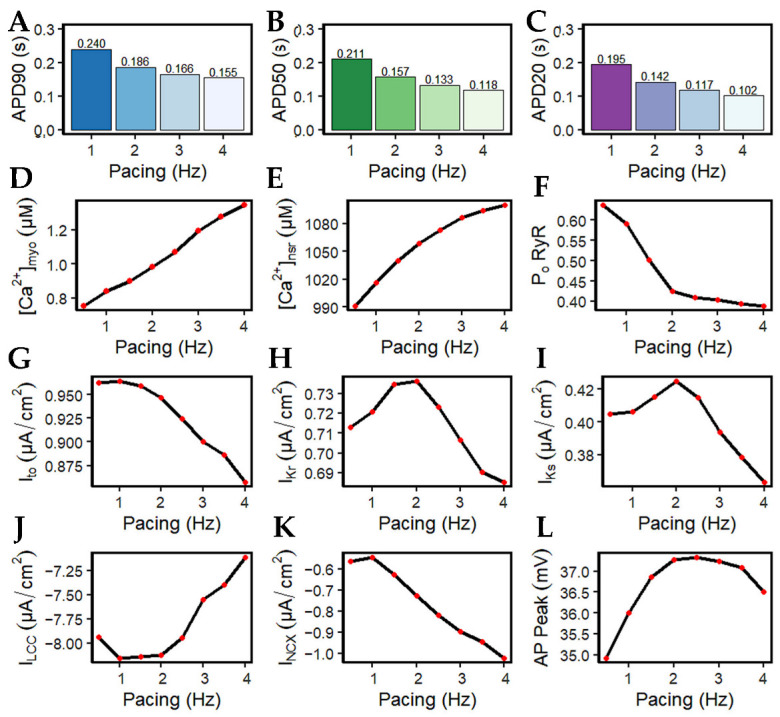
Frequency-dependent changes in AP morphology and its relationship to Ca^2+^ handling. (**A**–**C**) APD90, APD50, and APD20 decrease with higher pacing which indicates APD shortening due to frequent stimulus. (**D**) Peak Ca^2+^ transient increases during higher pacing. (**E**) SR Ca^2+^ loading denoted by maximum [Ca^2+^]_nsr_ levels at diastole. (**F**) Peak open probability of ryanodine receptor (RyR2) decreases with high pacing due to SR Ca^2+^ loading. (**G**–**I**) Rate-dependent changes in repolarizing K^+^ currents with the exception of I_K1_. (**J**) I_LCC_ density (minimum values; negative) decreases which also acts as negative feedback to Ca^2+^ influx. (**K**) Increasing depolarizing current (minimum values of tail current; negative) of I_NCX_ which represents a greater Ca^2+^ extrusion which shortens the APD. (**L**) Membrane potential amplitude peaks at 2.5 Hz (~37.5 mV) and begins to decrease at 3 Hz.

**Figure 9 biomolecules-13-01259-f009:**
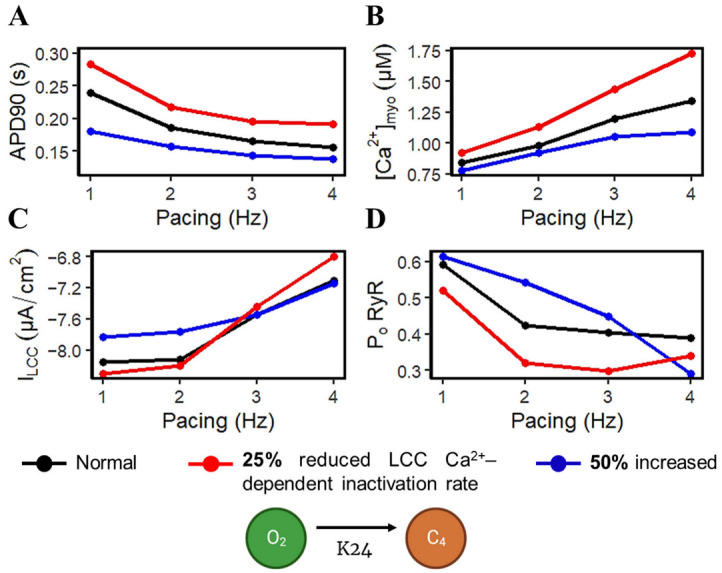
Varying l-Type calcium channel Ca^2+^–dependent inactivation rate. (**A**) APD90s at normal (black), 25% reduced (red), and 50% increased (blue) Ca^2+^–dependent inactivation rate effects. APD90s at increased LCC inactivation rate per pacing frequency is shorter than normal and reduced inactivation. (**B**) Peak [Ca^2+^]_myo_ still increases during higher pacing, but the 50% increased inactivation rate exhibits lower myoplasmic calcium concentrations. (**C**) I_LCC_ peak densities are varied, but reduced LCC inactivation rate at 3Hz and beyond shows a lower I_LCC_ peak. (**D**) RyR2 opening is higher during increased LCC inactivation rate.

**Figure 10 biomolecules-13-01259-f010:**
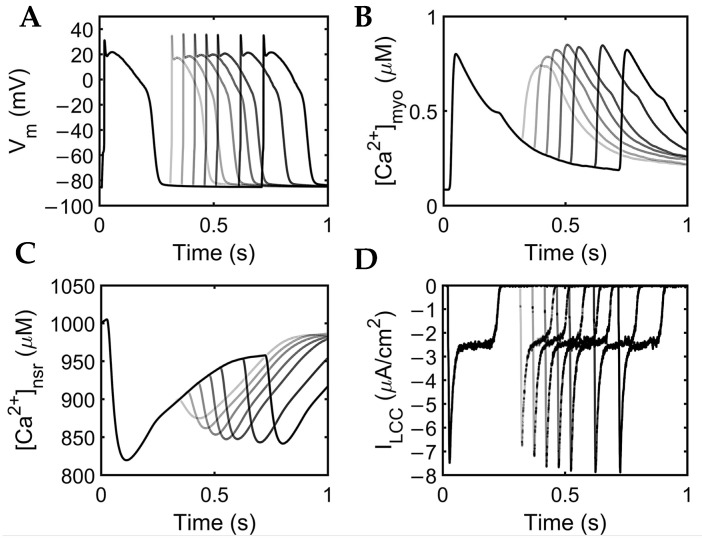
S1S2 restitution. (**A**) Action potential trains from 0.02 to 0.42 applied stimulus. (**B**) Calcium transient gradually rises with longer DIs. (**C**) SR Ca^2+^ initially displays a low amount of release at short DIs, which is also seen in a lesser Ca^2+^ concentration in the myoplasm ([Ca^2+^]_myo_). (**D**) l-type calcium channel exhibits low density at shorter intervals but returns to initial density due to a more complete SR Ca^2+^ recovery at longer intervals.

**Figure 11 biomolecules-13-01259-f011:**
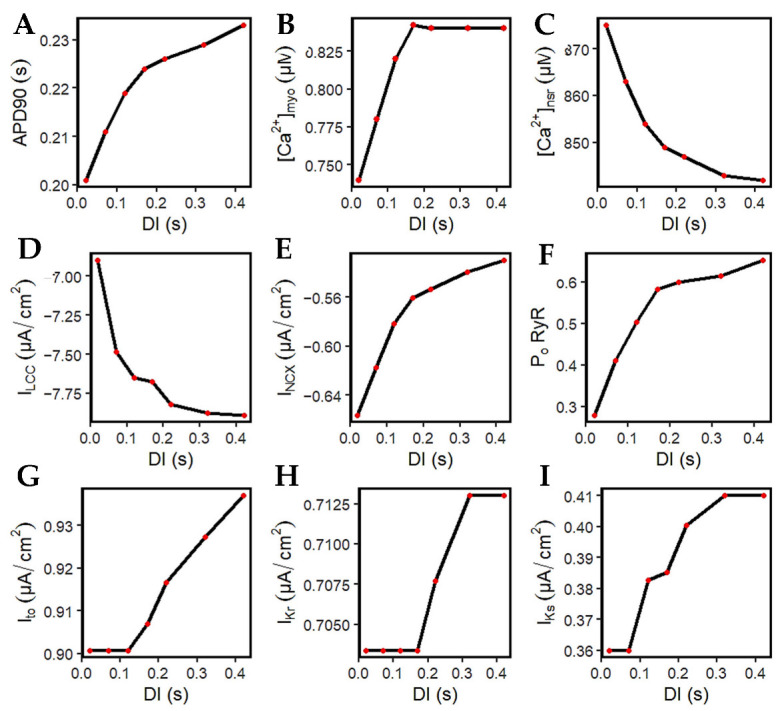
Mechanisms of APD restitution through S1S2 experiments. (**A**–**D**) Summary of amplitude changes in Figure 10. (**B**) Calcium transient amplitude reaches its proper levels at longer DIs (>0.17 s after initial stimulus). (**C**) Ca^2+^ release from the SR at shorter DIs are decreased. (**D**) I_LCC_ amplitude increases at longer DIs. (**E**) I_NCX_ depolarizing tail current decreases (becomes less negative) which allows for longer APD. (**F**) Open probability of ryanodine receptor increases with longer DIs due to higher SR Ca^2+^ release. (**G**,**H**) I_to_ and I_Kr_ at short DIs are at constant levels, then increase at longer intervals. (**I**) I_Ks_ increase partially affected by increasing [Ca^2+^]_i_ due to its calcium dependence.

**Figure 12 biomolecules-13-01259-f012:**
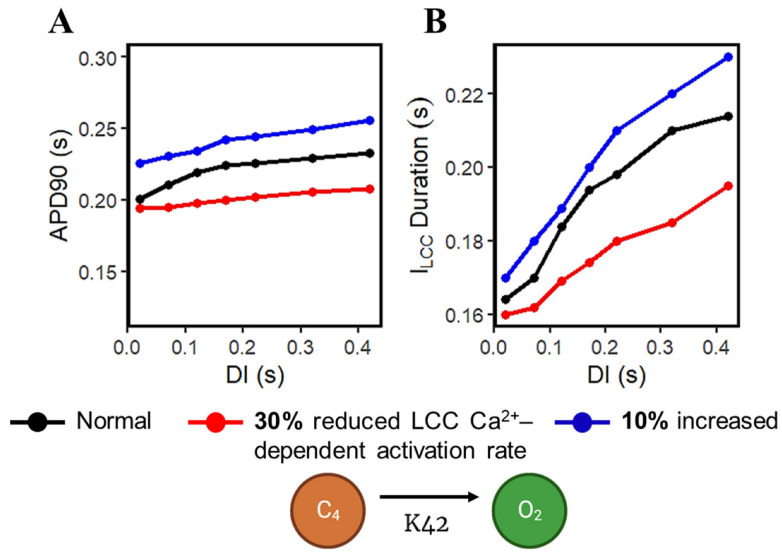
APD mediation by L-type calcium channel activation rate. (**A**) APD90s from 0.02 to 0.42 applied S2 stimulus exhibit upward trends. A 10% increased LCC calcium-dependent activation rate shows longer APD90 among all instances of increasing DIs. (**B**) I_LCC_ opening duration in seconds shows similar trend in all cases of APD90s per diastolic interval.

**Figure 13 biomolecules-13-01259-f013:**
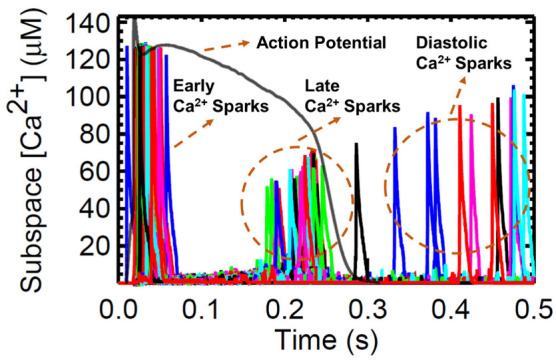
Representative sample (1.0% of 20,000 CRUs) of detected calcium sparks in the local subspace from 20 CRUs. Superimposed action potential depicting relationship between AP and Ca^2+^ sparks displays Ca^2+^ spark behavior in a normally functioning human AP. Minimal Ca^2+^ leak was observed in the late phase of the AP during transport of Ca^2+^ ions in cell repolarization.

**Figure 14 biomolecules-13-01259-f014:**
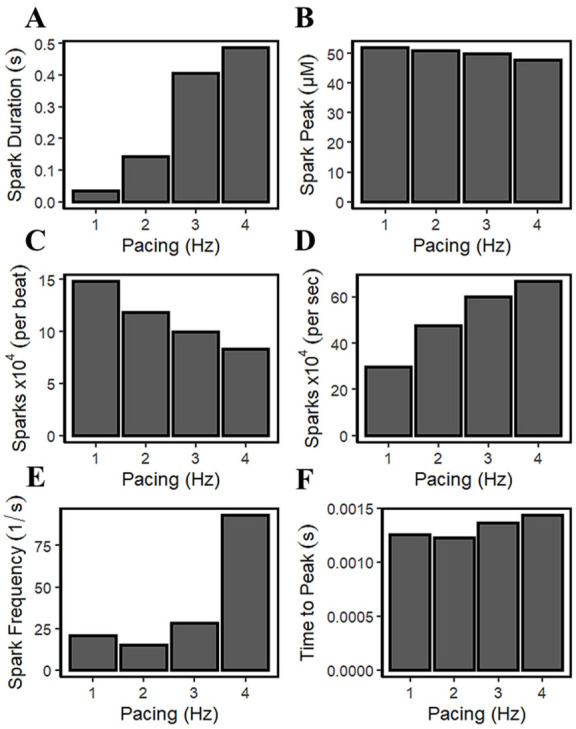
Spark characteristics with increasing frequency. (**A**) Spark duration shows an increase during higher pacing corresponding to the decrease in RyR2 open probability. (**B**) Average spark peak gradually declines. (**C**) 10,000 sparks per beat. Shorter window times represent the lower number of sparks observed in APD shortening in higher frequencies. (**D**) 10,000 sparks per second—more sparks are observed per second due to more frequent stimuli. (**E**) Spark frequency increase due to SR Ca^2+^ loading at faster rates. (**F**) Time-to-peak behavior of sparks with more frequent beats. The increasing number of sparks is primarily due to SR Ca^2+^ loading.

## Data Availability

Model code is available with the Appendix A.

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
