# Peer review of "Local Control Model of a Human Ventricular Myocyte: An Exploration of Frequency-Dependent Changes and Calcium Sparks"

_biomolecules, 2023, doi:10.3390/biom13081259_

Round 1

Reviewer 1 Report

In this study, the authors developed human ventricular action potential model with intracellular Ca cycling described by individual CRUs (or sparks). Some of the results are compared with experimental data. By incorporating the individual CRUs into the model, it is a step further from the exiting human ventricular action potential modes, and thus it is worth for publication. The following are my comments for revision.

1.       CRU model. In Fig.1, I only see one CRU, but the legend and the main text, it says two CRUs. Can you clarify this? It is also not to me how the CRUs are “spatially” arranged and coupled in the model. Is it the same as in Rice et al or as in Nivala et al in which Ca waves are allowed?

2.       Numerical Methods. The authors stated that ultra-fast Monte-Carlo method was used, which is a patented method. As the author may also know very well,  the so-called Gillespie method is widely used, which is an accurate algorithm and much faster than the traditional Monte-Carlo method. The question is: how much faster is the patented algorithm than the Gillespie algorithm? If it is much faster, then the authors should open algorithm free to the readers who want to simulate this current model in their future studies. If it is comparable, it should be clarified so that others can use the Gillespie method instead.

3.       Fig.13. It is interesting to see the late sparks. Do late sparks only occur around phase-3 of the action potential?  Are they caused by the recovery of RyRs and refilling of SR, or also related to the voltage? If they are caused by recovery of RyRs and SR, it should also occur much later, such as around 0.4 to 0.5 s in the figure.

4.       DAD and triggered activity. A major common issue of the previous human ventricular action potential models is that they are incapable of generating DADs and triggered activity due to their instability to generate spontaneous Ca releases and Ca wave. The advantage of incorporating thousands of CRUs into the model allows the model to exhibit spontaneous Ca release and thus DADs. I think that this is very important feature of detail Ca cycling models. This has been simulated in the rabbit models by Zhen Song et al in a number of studies. I am wondering if you overload this model (or tune the model with the conditions of heart failure), can this model generate DADs and triggered activity?

Author Response

The response is attached. 

Reviewer 2 Report

In this manuscript Alvarez and colleagues developed a ventricular cardiomyocyte human model that takes into account the stochastic behaviour of Ca2+ propagation within the cell. The model, that is an updated version of pre-existing models, includes several components of the excitation-contraction coupling like ionic currents, RyR2 stochastic behaviour, LTCC activation and inactivation properties, fluxes of Ca2+ within the sarcolemma etc.

I believe that this upgrade version provides important additions taking into account adaptations to mechanism of restitution, changes in frequency, Ca2+ leak from the reticulum.  More in detail the model ricapitulates, among the others: i) the stochastic nature nature of Ca2+ propagation via the Ca2+-release units. ii) the changes of Ca2+ propagation during the staircase protocol comprehensive of LTCC cynetics and Ca2+ sparks generation iii) the effect of FFR providing additional insight into the overall intracellular Ca2+ dynamics. The manuscript is clear, well written and suitable for publication upon minor adjustments.

The first paragraph of the section 3. Results is quite reduntant and similar to the abstract. I recommend to the authors to rewrite the section (or delete it).

Is the model taking into account of the contribution of store-operated Ca2+ entry? Is not a pivotal pathway in excitable cells, but seems to play a key role in condition of ER instability (like CPVT for example).

Reviewer 3 Report

Report on Local Control Model of a Human Ventricular Myocyte: An Exploration of Frequency-Dependent Changes and Calcium Sparks

The paper presents a stochastic human ventricular cardiomyocyte model for analyzing calcium (Ca2+) propagation in the heart cells. The model adapts to intracellular calcium dynamics, frequency-dependent changes, and spark regulation. Incorporating 20,000 calcium-release units (CRUs), it captures the calcium propagation behavior in cardiomyocytes, as well as the adaptability of the ryanodine receptors (RyRs) under different pacing frequencies. The model demonstrates the influence of L-type calcium channels and RyRs on action potential duration, force generation, and Ca2+ release during various pacing rates. Thus, it offers insights into calcium-related cellular processes and responses in the heart.

While the manuscript demonstrates commendable writing quality and a solid modeling approach, its current length is rather excessive, necessitating significant condensation for improved readability. The Methods and Results sections encroach upon territory typically reserved for the Discussion section, an issue that requires rectification. Extraneous discussions should be condensed within these sections, or relocated to the Discussion section. Moreover, the transparency of the subsection headers leaves room for improvement. To facilitate readers in locating pertinent results, these headings should provide a more unambiguous portrayal of the associated findings.

Major concerns:

1.     The abstract of the paper currently lacks clarity, imparting an impression of a multifaceted, somewhat directionless endeavor. It necessitates a thorough rewrite in order to function as a precise, concise summary, specifically delineating the authors' objectives.

2.     The model is constructed by using well known elements from other models and then new features are added. In order to for the reader to be able to comprehend the complete model it needs to be presented in its full form with all the equations. This can be done in supplementary material.

3.     The numerical method used to solve the equations should also be described in technical terms and not only with words. All the results of the paper are based on simulations and then the reader should now how the equations are solved. Again, this can be done in supplementary information.

4.     The code must be made publicly available (that seems to be the authors intention).

5.     The graphs in the paper appears to in very low resolution. Can that be improved?

6.     There are a number of minor improvements in this model compared to others, but there are few direct comparisons of the effect of individual changes. Again, figures in the supplementary where effects of new formulations would be helpful.

7.     Figure 1 is very generic an does not illustrate the present model with 20,000 CRUs very well.

8.     The CICR process is modeled in a very detailed manner, but the effect in terms of force is modeled by the simple formula (9) which I guess is phenomenological. Why is not the widely accepted model in [1] or the newer [2] used to accurately compute the forces?

9.     Another question regarding the balance of the accuracy of the model: The CICR process is given high priority but the essential effect of high gain and graded release can be obtained using much simpler models (see e.g [3]). Is the computational complexity of 20,000 CRUs justified?

Minor comments:

1.     I do not understand this statement: New findings from the O’Hara-Rudy model using non-diseased human mRNA and protein data made the modeling for different transmural cell types possible.

2.     The first sentence of the Discussion is also difficult to understand: Decades of cardiac modeling development have propelled this research into much closer human implications as accurate as possible.

[1]: https://www.cell.com/fulltext/S0006-3495(08)78384-X

[2]:https://www.sciencedirect.com/science/article/pii/S0022282817300639?casa_token=QkUEqRbkTAwAAAAA:quAV4yuLY2pGL_fTDYJrkTEQKuHLtr2v7D72Q9CXhqaVkdH2A5hOGmoWF14y0Z_ILAx-d9Rm6Q

[3]: https://doi.org/10.3389/fphar.2019.01648

The language is fine except for a few sentences mentioned in my report.

Round 2

Reviewer 1 Report

My comments are properly addressed except the one on DADs. In Song's studies (such the 2017 PNAS paper and the 2018 JMCC paper), large spontaneous Ca release induced DADs, which caused triggered activity, are shown.  The authors should mention that the current model does not show DADs and triggered activity as a limitation and clarify that they will be studying this using (or improving) this model in their future work.  
